# *Staphylococcus aureus* Pathogenicity in Cystic Fibrosis Patients—Results from an Observational Prospective Multicenter Study Concerning Virulence Genes, Phylogeny, and Gene Plasticity

**DOI:** 10.3390/toxins12050279

**Published:** 2020-04-26

**Authors:** Jonas Lange, Kathrin Heidenreich, Katharina Higelin, Kristina Dyck, Vanessa Marx, Christian Reichel, Willem van Wamel, Martijn den Reijer, Dennis Görlich, Barbara C. Kahl

**Affiliations:** 1Institute of Medical Microbiology, University Hospital Münster, 48149 Münster, Germany; jonaslange@yahoo.de (J.L.); Kathrin.Heidenreich@gmx.de (K.H.); katharinahigelin@gmx.de (K.H.); kristinadyck@gmx.de (K.D.); vani-marx@web.de (V.M.); christian.a.reichel@online.de (C.R.); 2Department of Medical Microbiology and Infectious Diseases, Erasmus Medical Center Rotterdam, 3015 CN Rotterdam, The Netherlands; w.vanwamel@erasmusmc.nl (W.v.W.); p.m.reijer@hotmail.com (M.d.R.); 3Institute of Biostatistics and Clinical Research, University Hospital Münster, 48149 Münster, Germany; Dennis.Goerlich@ukmuenster.de

**Keywords:** *Staphylococcus aureus*, cystic fibrosis, virulence genes, phylogeny, horizontal gene transfer

## Abstract

*Staphylococcus aureus* and cystic fibrosis (CF) are closely interlinked. To date, however, the impact of *S. aureus* culture in CF airways on lung function and disease progression has only been elucidated to a limited degree. This analysis aims to identify bacterial factors associated to clinical deterioration. Data were collected during an observational prospective multi-center study following 195 patients from 17 centers. The average follow-up time was 80 weeks. *S. aureus* isolates (n = 3180) were scanned for the presence of 25 virulence genes and *agr*-types using single and multiplex PCR. The presence of specific virulence genes was not associated to clinical deterioration. For the *agr*-types 1 and 4, however, a link to the subjects’ clinical status became evident. Furthermore, a significant longitudinal decrease in the virulence gene quantity was observed. Analyses of the plasticity of the virulence genes revealed significantly increased plasticity rates in the presence of environmental stress. The results suggest that the phylogenetic background defines *S. aureus* pathogenicity rather than specific virulence genes. The longitudinal loss of virulence genes most likely reflects the adaptation process directed towards a persistent and colonizing rather than infecting lifestyle.

## 1. Introduction

Cystic fibrosis (CF) is the most prevalent life-shortening genetic disorder in the Caucasian population, characterized by increased viscosity of exocrine gland secretions, congestion of physiological outflow pathways, and consecutive cystic and fibrotic remodeling of the respective organs [1,2]. Chronic and recurrent bacterial airway infections are the main cause of morbidity and mortality among CF patients [3].

*Staphylococcus aureus* is one of the first pathogens that CF patients encounter and the bacterium with the highest prevalence in the CF population [4,5]. In most cases, a long-term colonization with individual *S. aureus* clones is observed [4]. Persistent *S. aureus* clones tend to form a genetically and particularly phenotypically heterogeneous population with *S. aureus* acquisition most likely outside the hospital with only minor transmission events [6,7,8]. The genetical diversity is fostered by mutations and horizontal gene transfer with persistence of dominant lineages and a continuous process of intra-host adaptation [6,9]. Both geno- and phenotypic changes occur in a time-dependent manner and accumulate [10,11]. The phenotypic heterogeneity of *S. aureus* populations in CF patients is intensified by differences in gene expression, mediated by regulatory mechanisms responding to specific milieu conditions [6]. The overall trend of *S. aureus* adaptation in CF airways is directed towards a persistent and colonizing rather than infecting lifestyle [6,12]. Even in the case of exacerbation, the presence of *S. aureus* is restricted to the airways and systemic disease is rarely observed [6]. The preponderance of colonizing clones results from the suppression of virulence genes for the sake of an intensified expression of protective factors [6]. Manifold strategies have been described to evade immune response and antibiotic treatment, including biofilm formation [13,14] emergence of small colony variants (SCVs) [15,16,17], intracellular persistence [18], and *agr*-dysfunction [19], among others [14,20,21,22].

In children, the colonization with *S. aureus* is associated with increased inflammatory activity in the airways and a worse nutritional status [23,24,25]. For older patients, however, the impact of *S. aureus* carriage on lung function and disease progression has not yet been elucidated [26]. This deficit in knowledge is reflected by the absence of internationally accepted therapeutic guidelines concerning *S. aureus* treatment in CF patients [27,28].

In a prospective longitudinal multicenter study, we recently identified the presence of SCVs, high bacterial density in throat cultures, and elevated IL-6 levels as independent risk factors among others for worse lung function in patients who were older than 6 years [26]. Further analyses revealed associations between antibiotic treatment, age, and *S. aureus* clonal carriage profiles during persistence assessed by *spa*-sequence typing of almost 4000 *S. aureus* isolates from 183 CF patients during a 21-month prospective study [29]. Patients receiving intensive antibiotic therapy regimens more likely carried *S. aureus* isolates during persistence, which belonged to only one *spa*-type [7], while older patients carried isolates with related *spa*-types with mutations of the repeat region including deletions, duplications of repeats or point mutations within repeats, or dominant *spa*-types cultured [7].

The present analysis continues the evaluation of the abovementioned study intending to identify genetic bacterial factors associated to clinical deterioration. We hypothesized that either the presence of specific *S. aureus* virulence genes or the occurrence of certain phylogenetic backgrounds might be associated with a deterioration of the patients’ clinical status. To test our hypothesis, we scanned every isolate for the presence of 25 virulence genes (*cap 5/8*, *chp*, *cna*, *clf A/B*, *sdrC/D/E*, *fnbA/B*, *sasG/H*, *eap*, *emp*, *sea–sej*, *eta/etb*, *tst-1*, *pvl*, and *hlg*) and the *agr*-specificity groups 1–4, using mono and/or multiplex PCR. Furthermore, the host response to 19 of these virulence genes was assessed by measuring IgG levels against the specific proteins by a bead-based flow cytometry technique (xMAP1; Luminex Corporation, Austin, TX, USA) as previously described [30,31]. 

## 2. Results

### 2.1. Demographics and Statistics

The CF patients participating in the study on average were younger than the German CF population and showed better lung function [26]. Furthermore, there was an overrepresentation of nonF508del genotypes and male sex [26]. Fifty-eight patients (30.5%) suffered an exacerbation during the study period, whereas the clinical status of the remaining 132 (69.5%) subjects did not deteriorate considerably. Subjects suffering one or more exacerbations showed a poorer lung function (mean 75% FEV_1_% predicted) than healthier patients (mean 84% FEV_1_% predicted) (*p* = 0.034 *, Mann–Whitney U test). In total, 3180 *S. aureus* isolates were analyzed (1312 from throat cultures, 1065 from nose cultures, and 803 from sputum cultures). On average, 3.22 (range: 1–12) different *S. aureus* clones as determined by *spa*-typing (Appendix A were retrieved from the subjects’ airways. The majority of patients (108; 57.8%) carried at least one dominant clone that was detected on every outpatient clinic visit. Six *spa*-types were most prevalent within this *S. aureus* population belonging to the *spa*-types t084, t012, t091, t015, and t002 (Appendix A).

### 2.2. Quantity of Virulence Genes and Clinical Parameters

The starting point for this analysis was the hypothesis that a broad virulence gene profile might be associated to a deterioration of the patients’ clinical parameters. However, there was no statistical difference in mean virulence gene quantity between patients suffering from an exacerbation during the study period (mean of 14.09 virulence genes) and subjects not witnessing exacerbations (mean of 14.29 virulence genes).

### 2.3. Time-Dependent Quantity of Virulence Genes

The number of virulence genes present in the *S. aureus* clones populating the patients’ airways was documented at every outpatient visit, enabling a longitudinal analysis concerning time-dependent changes in the quantity of virulence genes exhibited by the patients’ *S. aureus* clones. At the beginning of the study period, the patients’ *S. aureus* clones carried on average 14.51 of the 25 virulence genes, whereas at the end, a mean of 14.17 virulence genes was present. A generalized estimating equation model was applied using Poisson distribution with log link function. Appropriate model terms for repeating observations were included using an independent working correlation matrix. Within the model, we could show a significant decrease in virulence gene quantity (*p* = 0.029 *, Figure 1) over time. On average, patients lost $5.6\times 10^{-4}$ genes per day. Extrapolated (beyond the maximal follow-up of this study), this would translate to a loss of one gene every five years (exactly, 1.022/5 years).

### 2.4. Specific Virulence Genes and Clinical Parameters

The next step consisted in the analysis of a possible association between single virulence genes and clinical parameters. Therefore, the mean prevalence counts of the individual virulence genes in different patient subgroups were compared. Patients suffering an exacerbation during the study period (n = 58) showed similar prevalence counts of the different virulence genes compared to healthier subjects (n = 132) (Figure 2a–c). The only exception was patients with isolates belonging to *agr*-type 4, showing a significant overrepresentation among patients with an exacerbation (2.34% vs. 7.72%) (*p* = 0.008 **, Mann–Whitney U test, Figure 2c). *Agr*-type 1 showed a considerable underrepresentation among patients witnessing an exacerbation (*p* = 0.051, Mann–Whitney U test). To narrow the spectrum of considered subjects to patients with the presumably most severe disease, patients with intensive antibiotic therapy and exacerbation (n = 35) were compared to the other subjects. In this subgroup, isolates belonging to *agr*-type 1 were significantly underrepresented (*p* = 0.011 *, Mann–Whitney U test, Figure 2d).

### 2.5. Complementary Data Concerning a Humoral Response Towards S. aureus Antigens

*S. aureus* toxin directed antibody levels in CF patients have been associated to the severity of exacerbations and decline in FEV_1_% [32]. Therefore, we measured IgG levels against 19 virulence factors in serum samples of 182 subjects and compared them to antibody levels of 53 healthy adults with persistent nasal *S. aureus* colonization (Appendix A). While IgG levels against six antigens were significantly elevated in patients compared to controls (Table 1), IgG levels against seven virulence factors were significantly associated to a decline in FEV_1_% predicted (Table 1).

### 2.6. Phylogeny and Virulence Genes

In the preceding analyses, it became obvious that neither the number of virulence genes present nor the presence of specific virulence genes was associated to changes in the patients´ clinical parameters. Nevertheless, the significant differences of the mean prevalence counts of the *agr*-types 1 and 4 among patients with an exacerbation suggest an influence of the phylogenetic background on varying degrees of *S. aureus* pathogenicity in CF patients. To further pursue these hints, we tested for a link between agr-type and phylogeny.

*Agr*-type and virulence gene profile: Clones, which belonged to varying *agr*-types, differed significantly concerning the counts of virulence genes (*p* = 0.001 ***, ANOVA, Appendix A). *Agr*-type 4 isolates carried less virulence genes compared to isolates belonging to *agr*-type 1. Next, the prevalence counts of the specific virulence genes in patients whose *S. aureus* clones predominantly belonged to one of the four *agr*-types were compared. In this analysis, clear and significant differences in almost all considered virulence genes were found (Kruskal–Wallis test, Figure 3a). Benjamini and Hochberg (False discovery rate) correction was applied to adjust for multiple testing. In a final analysis, it was tested for associations between the most prevalent *spa*- and *agr*-types. This revealed that each of the most prevalent *spa*-types could be clearly assigned to one *agr*-type (*p* < 0.001, Chi-squared test, Figure 3b).

*Agr*-type and longitudinal virulence gene quantity: In our preceding analysis, we showed that the counts of virulence genes present in the patients’ *S. aureus* clones demonstrated a time-dependent decrease (Figure 1). Reconducting this analysis and taking into account the *agr*-type of the patients’ isolates revealed clear and significant differences in the process of time-dependent alterations of the virulence gene quantity between isolates belonging to the different *agr*-types (*p* = 0.0083 **, Wald test). Isolates belonging to *agr*-type 1 and 3 showed a longitudinal decline in virulence gene quantity, whereas the virulence gene number in isolates belonging to *agr*-type 2 slightly increased and *agr*-type 4 isolates remained stable in virulence gene numbers (Figure 4).

### 2.7. Cluster Analyses

Since the isolates belonging to the different *agr*-types significantly varied in the prevalence of many virulence genes (Figure 2), it was considered that certain combinations of virulence genes rather than single virulence genes might be associated to a deterioration in clinical parameters. To test this hypothesis, a distance matrix was created deploying Euclidian distance and then hierarchical clustering via ward linkage was applied. The virulence gene profiles were grouped into four association clusters, implying that the virulence gene profiles of patients within a cluster are more similar to each other than to virulence gene profiles of other clusters. In a second step, we analyzed for associations between these clusters and the different *agr*-types, rendering significant results (*p* = 0.001 ***, Chi-squared test, Figure 5). Next, it was investigated whether exacerbation rates differed significantly between the four virulence gene clusters, which was not the case.

### 2.8. Virulence Gene Plasticity and Environmental Stress

Environmental stress consisting of antibiotic therapy and/or immune response has been shown to be a powerful inducer of hgt, in particular of bacteriophage mediated transduction [33,34,35]. Since this process can entail clinical consequences and can be induced iatrogenically, the data were checked for hints pointing towards stress-induced virulence gene plasticity in the study population. The patients were divided using the “environmental stress high/low” parameter. Patients assigned to the “environmental stress high” group showed at least one exacerbation during the study period and received intensive antibiotic treatment; 633 *S. aureus* isolates of 35 patients were exposed to high amounts of environmental stress. There were significant differences in plasticity for three out of 25 tested virulence genes (*chp*, *eta*, *etb*), all of them showing increased plasticity in case of high environmental stress (Mann–Whitney U test, Figure 6).

## 3. Discussion

A prospective longitudinal multi-centre study consisting of 195 patients receiving treatment in 17 specialized CF centres was conducted to identify genetic bacterial factors associated to clinical deterioration in the patient cohort. It was tested whether the presence of specific *S. aureus* virulence genes or the occurrence of certain phylogenetic backgrounds were linked to differing clinical status of the subjects. Furthermore, the data were analysed for longitudinal changes in the *S. aureus* virulence gene composition and stress dependent virulence gene plasticity.

The analyses revealed several interesting findings that will be discussed in the following.

### 3.1. Specific Virulence Genes and Clinical Parameters

None of the virulence genes characteristic of *S. aureus* (*tst-1* and *lukS/F*, among others) was associated to exacerbations. This finding supports the hypothesis that, given no susceptible host, the presence of single virulence genes is not associated with a significant increase in the overall severity of *S. aureus* infections and, consequently, that a screening for such “super weapons” is not required [36,37,38,39]. These results, however, only apply to *S. aureus* disease in CF patients and neither could nor should be generalized. Moreover, there are significant regional differences in the prevalence counts of the various virulence genes. For instance, community-acquired methicillin-resistant *S. aureus* (CA-MRSA) strains, which mostly carry the Panton-Valentine leucocidin (PVL, *lukS/F*), are rare in Germany while widespread in the US [40,41,42]. The significant differences in the prevalence counts of the *agr*-types 1 and 4 between patients with and without exacerbations most likely did not arise from differing intrinsic gene regulatory activities of the varying *agr*-types, since the *agr*-system has been described to be inactive in CF patients most of the times [19,43]. Rather, the observation of widespread *agr*-inactivity or dysfunction in CF airways suggests a link between the *agr*-system and *S. aureus* phylogenetic background as an explanation for differing gene prevalence counts in exacerbating patients [39].

### 3.2. Phylogeny and Pathogenicity

The identification of the different *agr*-types as the only parameter that was associated to the subjects’ clinical status supports the hypothesis that the basic unit of *S. aureus* pathogenicity is not the presence of specific single virulence genes but most likely certain combinations of genes linked to a clone’s phylogenetic background [39,44]. The phylogenetic indicator function of the *agr*-system is highlighted by further analyses, revealing considerably varying virulence gene profiles between patients whose *S. aureus* clones differed concerning their *agr*-types (Figure 3a) and the observation that each prevalent *spa*-type could be clearly assigned to one *agr*-type (Figure 3b). The results of the cluster analyses added further coherence to these findings: the different *agr*-types showed a significantly unequal distribution between the four virulence gene clusters (Figure 5).

### 3.3. Longitudinal Decline in Virulence Gene Quantity

The analyses revealed a significant time-dependent decline in the number of virulence genes exhibited by the subjects’ *S. aureus* clones (Figure 1). The decrease may seem negligible at first glance; however, it has to be taken into account that this analysis is highly limited in two dimensions. On the one hand, it only covers a short cutout of the longstanding interaction between *S. aureus* and its human host. On the other hand, only a small fraction of the extensive *S. aureus* virulence genome is included. Thus, a hypothetical extrapolation concerning follow-up time and considered virulence genes would put the results into a different perspective. Hence, this finding is considered as a clear hint towards a longitudinal decline of *S. aureus* virulence gene quantity in CF patients and thus indicates that *S. aureus* shows not only high phenotypic diversity and increased degrees of gene plasticity in CF airways but also long-term changes in its (virulence) gene composition. The finding is also consistent with the persistent and colonizing rather than infectious lifestyle, which *S. aureus* displays in CF patients [6]. Similar processes have also been proven for chronic *P. aeruginosa* infection in CF patients [45,46]. The gradual loss of virulence genes could be beneficial for the bacterium concerning the evasion of host defenses, since many virulence factors are targets for the immune system as obvious by mounting of specific IgG against such virulence factors (Table 1) [45]. Future research has to show whether there is evidence for this trend on the level of a more comprehensive virulence gene profile analysis using whole genome sequencing (WGS). While Azarian et al. used WGS to investigate intra-host diversity of sequential MRSA isolates in four CF patients, the authors focused in their analysis, for which they used sequencing data of a mean of 21 isolates per patient from a mean period of 1403 days on the evidence of switched intergenic regions, allele frequency, and single nucleotide polymorphisms but not the presence or absence of special virulence genes [9]. Therefore, it is not possible to compare their data with our analysis. 

### 3.4. Virulence Gene Plasticity and Environmental Stress

In this analysis, the virulence gene plasticity was used as a surrogate parameter to estimate the potential for hgt. It is considered a reasonable parameter for this purpose since genome plasticity forms a necessary precondition for hgt. Furthermore, surroundings that are associated to increased rates of virulence gene plasticity in the data have also been identified as inductors of hgt [47,48]. The analyses revealed significantly increased plasticity rates for several genes in the presence of high environmental stress (Figure 6). These results are in line with research having identified environmental stress as an inducer of hgt and suggest an iatrogenic and host-related impact on the *S. aureus* virulence genome [34,35]. According to the “insurance hypothesis”, genomic plasticity is a necessary prerequisite for adaptation to changing and challenging surroundings [49]. That these surroundings themselves increase the potential for hgt is an important fact that treating physicians should be aware of. Particularly sub-inhibitory concentrations of antibiotics, which are often observed to occur in CF patients, have been shown to be powerful inducers of hgt [50].

### 3.5. Strengths and Limitations of the Study and Analyses

The study population differs from the German CF population concerning age, sex, lung function, and genotype [26]. The “No *P. aeruginosa*” inclusion criterion is considered to be the reason for the subjects’ young age, since with increasing life span patients become more prone to colonization and infection with *P. aeruginosa* [5]. The young age in turn presumably accounts for the subjects’ better lung function compared to the German CF population, since it is an age-correlated parameter characterized by a longitudinal decline [51,52]. The “No *P. aeruginosa*” criterion presumably also accounts for the increased prevalence of non-F508del genotypes since those genotypes have been shown to exhibit a delayed colonization by *P. aeruginosa* compared to F508del homozygous patients [53]. There are no clear hints that these demographic incongruities go along with a limited transferability of the findings to the basic CF population. Both groups show a similar annual decline in lung function and thus do not seem to differ substantially in the course of the disease [26,51,52]. A shortcoming of the study is the rather short follow-up time of app. 80 weeks, which only covers a narrow cutout of the long-lasting interaction between *S. aureus* and its human host. Moreover, the analyses are primarily based on genetic data. Complementary proteomic data would be desirable, since differences in gene expression contribute to the high level of phenotypic diversity among *S. aureus* clones [54]. Especially the diversity of the *S. aureus* exoproteome is considered to be relevantly shaped by gene regulation mechanisms [55,56].

## 4. Conclusions

The results of our prospective multicenter study suggest that the phylogenetic background defines *S. aureus* pathogenicity rather than specific virulence genes. The significantly increased gene plasticity rates in the presence of environmental stress suggest an iatrogenic and host influence on *S. aureus* genome plasticity. The longitudinal loss of virulence genes might reflect the adaptation process directed towards a persistent and colonizing rather than infecting behavior of *S. aureus* in CF patients. 

## 5. Material and Methods

### 5.1. Study Design

The data underlying this analysis were generated during an prospective multicenter study (ClinicalTrials.gov, NCT00669760). An ethical statement was obtained at the main study center with the central laboratory in Münster, Germany (2007-496-f-S) on the 1 June 2009. Written informed consent was obtained from all patients and from parents if patients were younger than 18 years. 

The study population consisted of 195 CF patients receiving treatment in 16 German and one Austrian CF center. CF patients older than six years whose airways were persistently colonized by *S. aureus* during the study period (more than 50% of specimen positive for *S. aureus*) were included into the trial. The follow-up time was on average 80 weeks, with a mean of seven outpatient clinic visits per patient, usually performed at an interval of three months. Inclusion criteria were the presence of a minimum of two airway cultures positive for *S. aureus* during an interval of six months in the year before study enrollment, and if more than four cultures were available, at least 50% of them had to harbor *S. aureus*. To prevent possible confounding caused by the presence of bacteria other than *S. aureus*, potential subjects showing specimens positive for *Pseudomonas aeruginosa* and/or *Burkholderia cepacia* complex (BCC) were to be ruled out a priori. At each outpatient visit, nasal and throat and/or sputum samples were retrieved and sent to the central study laboratory in Münster, where the material was cultivated following standardized procedures for CF airway specimens as described elsewhere [57]. Furthermore, antibiotic regimens and clinical parameters were monitored and documented longitudinally. For a more detailed description of the study, see Junge et al. [26]. The obtained *S. aureus* isolates (n = 3893) were distinguished deploying the *spa*-typing method [58]. The “Based upon repeat pattern” method (BURP, Ridom StaphType software) was used to analyze the clonal relatedness of different *spa*-types differentiating 266 *spa* types (Appendix A) [29,59]. In this analysis, every isolate was scanned for the presence of 25 virulence genes and the 4 *agr*-specificity groups, using single and multiplex PCR. 

In the present study, 3180 *S. aureus* isolates obtained from 190 patients were analysed; five subjects were excluded because no persistent *S. aureus* has been detected in their specimens; and 713 isolates of the former 3893 isolates [33] were disregarded due to identity of isolates in regard of the same visit, sample (nose or throat swab or sputum), *spa*-type, and virulence gene profile.

### 5.2. Virulence Gene Profiles

The obtained *S. aureus* isolates (n = 3180) were scanned for the presence of 25 virulence genes and the four *agr*-specificity groups (*cap 5/8*, *chip*, *cna*, *clf A/B*, *sdrC/D/E*, *fnbA/B*, *sasG/H*, *eap*, *emp*, *sea*–*sej*, *eta/etb*, *tst-1*, *pvl, hlg*, and *agr*), using single and multiplex PCRs as described elsewhere (Appendix A [60,61,62,63,64,65,66,67]).

### 5.3. Parameters

The study provided an extensive data set, consisting on the one hand of the patients’ clinical data and on the other hand of the results of the microbiological analyses. Since the aim of this analysis was to screen for correlations between these two different data domains, reasoned parameters were to be established to render the data accessible for statistical analysis, while at the same time, oversimplification had to be avoided to generate data that is of use for clinicians. The parameters were chosen and defined as follows:

#### 5.3.1. Exacerbation

The treating physicians monitored the patient’s clinical status according to the Fuchs criteria (change in sputum/sinus discharge, new or increased dyspnea/cough/hemoptysis/malaise/weight loss/sinus tenderness, temperature over 38 °C, radiographic changes indicative of lung infection, and decrease in pulmonary function by 10%) [68]. If at one visit four or more Fuchs criteria were met, the respective patient was assigned to the “exacerbation” group.

#### 5.3.2. Lung Function

Spirometric lung function testing was conducted on every outpatient clinic visit. Lung function was expressed as forced expiratory volume (1 s) (FEV_1_) percent of normal. Comparison with a healthy cohort was achieved by using the Quanjer prediction model [69]. During the study’s follow-up, lung function changes over time were negligible. Thus, the mean lung function (FEV_1_%) throughout all outpatient clinic visits was calculated for every patient.

#### 5.3.3. Virulence Gene Prevalence Counts

To analyze for associations between the presence of specific virulence genes and a patient’s clinical outcome, the calculation of individual virulence gene prevalence counts was necessary, which display the percentage of isolates in which the respective virulence gene was present throughout the study period.

#### 5.3.4. Quantity of Virulence Genes

In addition to a possible clinical impact of single virulence genes, it was also evaluated whether a broad or narrow spectrum of virulence genes in general correlated with changes in clinical outcomes. Thus, it was reviewed for every visit how many of the 25 virulence genes were present in the patients’ *S. aureus* clones. This count was conducted in a complementary way, meaning that, at every visit, each proven virulence gene was only counted once per clone. This parameter also enabled a longitudinal analysis of a possible time-dependent change in the number of virulence genes incorporated by the *S. aureus* clones populating the patients’ airways.

#### 5.3.5. *Agr*-Type

The accessory gene regulator (*agr*) not only represents one of *S. aureus’s* most important virulence regulators but also is tightly linked to the bacteria’s phylogenetic background [39,44,70,71]. To elicit possible relations between *agr*-type, virulence gene prevalence counts, and *spa*-types, the isolates of patients were analyzed in terms of dominance of a specific *agr*-type. To belong to the group with dominant *agr*-types, at least a two-third majority of patients’ isolates had to belong to one of the four *agr*-types, which was the case for 154 patients. Patients without a dominance of one *agr*-type (36 subjects) were excluded from further analyses regarding this parameter.

#### 5.3.6. Environmental Stress

Previous research has shown an impact of antibiotic therapy and host immune response on genomic variability and horizontal gene transfer (hgt) [33,35,50]. Thus, the parameter “environmental stress” was established, which includes all patients who suffered an exacerbation during the study period and received intensive antibiotic therapy. The latter was defined as ≥7 cycles of antibiotic therapy during the study period. The environmental stress parameter was used to compare gene plasticity and hence the potential for hgt in *S. aureus* isolates that were exposed to high amounts of environmental stress to isolates subsisting in a less challenging environment.

#### 5.3.7. Virulence Gene Plasticity and Horizontal Gene Transfer

To quantify the plasticity of the virulence genes, the virulence profiles of the patients’ *S. aureus* clones were compared from visit to visit. Events of gain and loss of virulence genes were counted, resulting in a value for every virulence gene of every clone of each subject, representing its variability in this specific clone of this specific patient. To receive a variability value for every virulence gene for every patient independent of the different clones, the cumulative sum of genetic switch events of each clone of the patient was calculated, resulting in a variability value for every virulence gene for each patient, which later was set into relation to other parameters. The virulence gene plasticity was used as a surrogate parameter to estimate the potential for hgt.

#### 5.3.8. Antistaphylococcal Antibodies

IgG antibody levels were quantified applying a bead-based flow cytometry technique (xMAP ^®^; Luminex Corporation) as previously described [30,31]. For further information, see Junge et al. [26].

### 5.4. Statistical Analyses

Study data were analyzed using descriptive and inductive statistical methods. For data description, absolute and relative frequencies were used for categorical variables while continuous parameters were described by mean and standard deviation or by median and interquartile range. To check for normal distribution, histogram plots and the Kolmogorov–Smirnov test were applied. For group comparisons, t-tests and ANOVA were applied. In the case of non-normal distributions nonparametric methods (e.g., Mann–Whitney U tests) were applied. Associations between categorical variables were tested with Chi-squared tests. Regression coefficients were tested using the Wald-test statistic. Further details of the applied statistical methods are given in line with the results of the presented analyses. 

The statistical analyses were conducted using SPSS (IBM Corp. Released 2017. IBM SPSS Statistics for Windows, Version 25.0. Armonk, NY, USA: IBM Corp.). The significance level for all performed statistical tests is 0.05. All statistical tests have to be considered exploratory. Results, thus, might need to be validated in prospective trials. Significant *p*-values, thus, indicate noticeable effects and not significant results in a confirmatory sense. For illustrations, significance levels are indicated by *, **, and ***.

## Figures and Tables

**Figure 1 toxins-12-00279-f001:**
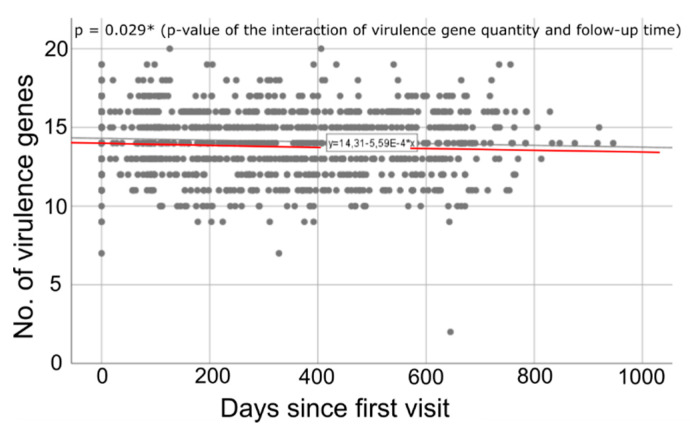
Time-dependent quantity of virulence-genes. The red line indicates the averaged loss of virulence genes over time for the study population. The grey dots indicate the virulence gene quantity of the individual *S. aureus* clones at different dates during the study. Significance levels are indicated by *, **, and ***.

**Figure 2 toxins-12-00279-f002:**
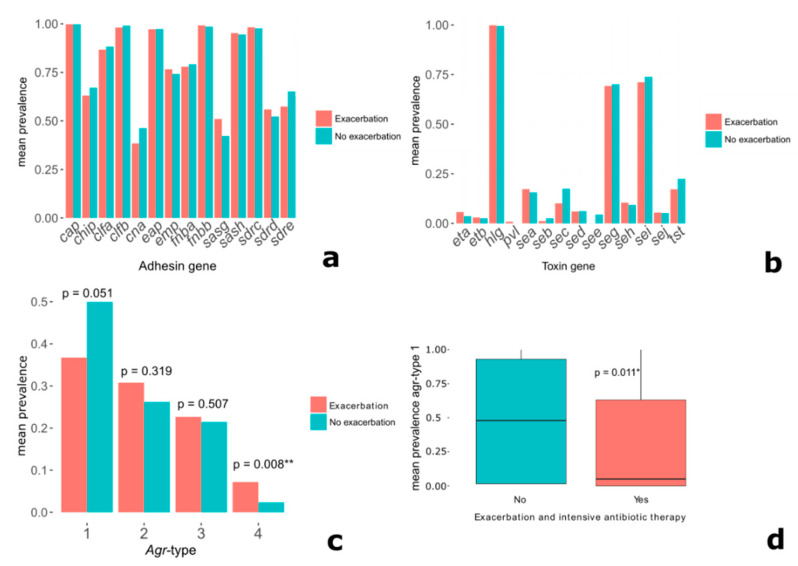
Prevalence of virulence genes and *agr*-types in the investigated CF *S. aureus* isolates. (**a**) Prevalence of *cap*, *chip*, and adhesin-genes in exacerbation vs. no-exacerbation patients; patients without exacerbation are indicated by blue bars, and patients with exacerbation are indicated by red bars. (**b**) Prevalence of toxin-genes in exacerbation vs. no-exacerbation patients; (**c**) prevalence of *agr*-types in exacerbation vs. no-exacerbation patients (Mann–Whitney U test); and (**d**) prevalence of the *agr*-type 1 among subjects without exacerbation compared to exacerbation patients who also received intensive antibiotic treatment (Mann–Whitney U test). Significance levels are indicated by *, **, and ***.

**Figure 3 toxins-12-00279-f003:**
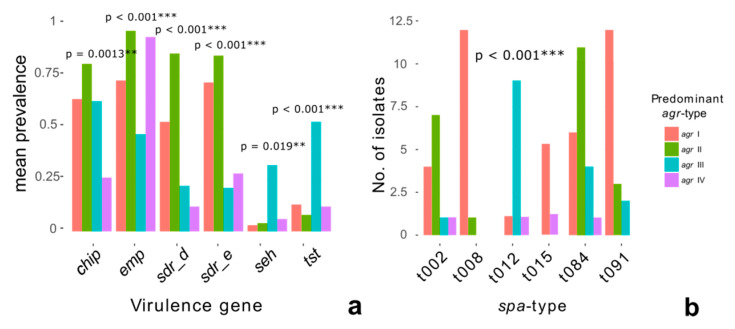
Association of *agr*-types and virulence genes or *spa*-types. (**a**) Association between *agr*-types and virulence gene prevalence counts (Kruskal–Wallis test); the 4 *agr*-types are indicated as follows: 1 = red, 2 = green, 3 = blue, and 4 = purple. (**b**) Association between *agr*-types and prevalent *spa*-types (Chi-squared test). Significance levels are indicated by *, **, and ***.

**Figure 4 toxins-12-00279-f004:**
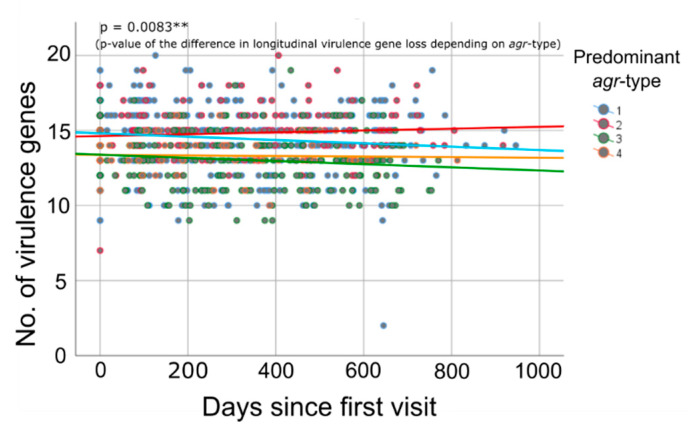
Longitudinal development of virulence gene counts depending on *agr*-type (Wald Test)**.** Colored lines indicate the averaged loss of virulence genes over time for patient groups showing a predominance of an *agr*-type (blue = *agr*-type 1, red = *agr*-type 2, green = *agr* type 3, and yellow = *agr*-type 4). The colored dots indicate the virulence gene quantity of the individual *S. aureus* clones at different dates during the study. Significance levels are indicated by *, **, and ***.

**Figure 5 toxins-12-00279-f005:**
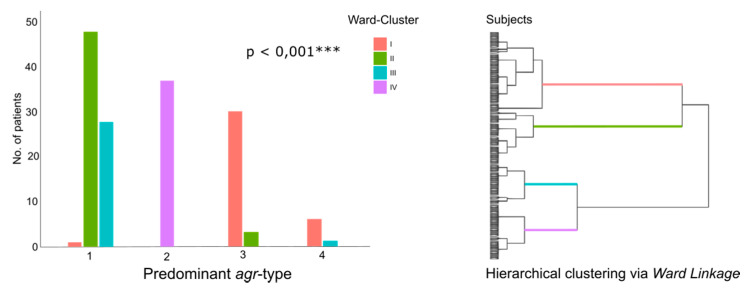
Association between the different *agr*-types and the four virulence gene clusters (Chi-squared test; red = cluster 1, green = cluster 2, blue = cluster 3, and purple = cluster 4). Significance levels are indicated by *, **, and ***.

**Figure 6 toxins-12-00279-f006:**
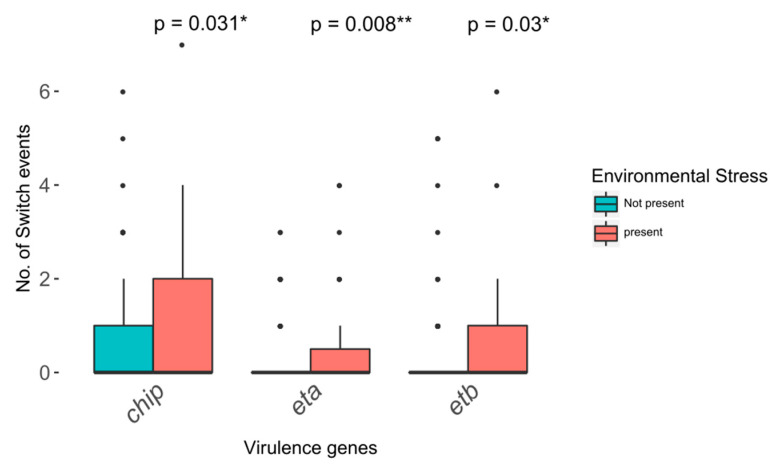
Plasticity of virulence genes depending on the presence of environmental stress (Mann–Whitney U test). Events of loss and gain of virulence genes in isolates exposed to environmental stress are indicated in red, and genetical switch events occurring in isolates residing in less-challenging environments are indicated in blue. Significance levels are indicated by *, **, and ***.

**Table 1 toxins-12-00279-t001:** Significant IgG levels against *S. aureus* antigens in patients vs. healthy controls and estimated effect on forced expiratory volume (1 s) FEV_1_% ^a^.

Antigen	Mean IgG Level All Patients (±SE) ^b^	Mean IgG Level Controls (±SE) ^b^	*p* Value ^c,d^	Estimated Effect on FEV1% ^e^	*p* Value ^d^
**CHIPS**	11492 (±220)	11019 (±349)	0.104	−0.00073	0.0185
**ClfB**	4587 (±222)	4092 (±338)	0.3809	0.002079	<0.0001
**Eta**	3999 (±447)	2043 (±424)	0.1180	−0.00061	0.0033
**Etb**	613 (±113)	320 (±94)	0.0032	−0.00700	<0.0001
**HlgB**	13621 (±174)	9878 (±407)	<0.0001	−0.00109	0.0088
**LukF**	4079 (±133)	2718 (±243)	<0.0001	−0.00305	<0.0001
**LukS**	14097 (±133)	7134 (±524)	<0.0001	−0.00169	<0.0001
**SdrE**	3139 (±201)	1992 (±221)	0.0253	−0.00179	<0.0001
**SED**	1106 (±118)	1292 (±268)	0.0498	0.001490	0.0377
**SEH**	2290 (±291)	2174 (±359)	0.0113	−0.00047	0.0986

^a^ Data are partially retrieved from our earlier study [26]; ^b^ significant difference of results between 182 patients and 53 healthy controls; ^c^
*p* values of differences between patient and controls groups (Mann–Whitney U test); ^d^ adjusted *p*-values (Bonferroni correction); and ^e^ IgG levels are modelled as continuous factors. Estimated effects are therefore interpreted as mean change in FEV1% predicted.

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
