# Peer review of "Staphylococcus aureus Pathogenicity in Cystic Fibrosis Patients—Results from an Observational Prospective Multicenter Study Concerning Virulence Genes, Phylogeny, and Gene Plasticity"

_toxins, 2020, doi:10.3390/toxins12050279_

Round 1
Reviewer 1 Report
Lines 59-64: The hypothesis centers upon the presence of virulence factors carried by the S. aureus. Of course, the other biology in pathogenesis is the immune status of the host. The study would be far stronger if one also had basic information on the immune response of the patients, e.g., presence or abscence of anti-toxin antibodies against the toxins expressed in the strains, then correlated with progression of lung dysfunction over time. The reviewer understands that this is not what the investigators proposed to study, but it is the study that the readers want to see, especially as readers of the journal “Toxins”.
Lines 189 - 192: While the loss of one gene every 5 years is statistically significant, one wonders if this is biologically signifcant. It seems unlikely as the patients as gene loss did not correlate with disease progression.
Line 284: It appears that presence of seb in Figure 2b did show increased exacerbation. This gene has also been found in worse oucome with S. auerus bacteremia.
Line 335: There are two other limitations that should be highlighted. First, the study of the bacteria relies solely upon gene presence and not gene expression. This is a major limitation as S. aureus gene expression in vivo has been emphasized in CF lung that does not correlate with gene presence (see Goerke et al. Infect Immun 68: 1304, 2000; Riquelme et al. Front Immunol 11: 385, 2020). Second, this study focuses upon the bacterium, but the host response is not included. Humans show variable responses to making antibodies to toxins, e.g., some women failed to make antibodies against tst, and they remained susceptible to toxic shock syndrome even after recovering from 1 episode.
Minor issues:
Line 18: The sentence construction causes difficulty in understanding. Specifically, the phrase “hypothesis that rather the…” appears, but it does not work in English grammar and is very unclear…..This needs to be revised.
Line 42: “infecting pheyotype” is unclear and needs to be defined, although it has been used previously, it is a term that is not in wide use. An earlier paper should be referenced that discusses the different types of S. aureus in the CF lung (Goerke C et al. Environ Microbiol. 9 :3134, 2007)
Line 56: The gramar in this sentence is difficult, making it hard to understand. This needs to be revised.
Author Response
Reviewer 1
Reviewer’s comment: Lines 59-64: The hypothesis centres upon the presence of virulence factors carried by the S. aureus. Of course, the other biology in pathogenesis is the immune status of the host. The study would be far stronger if one also had basic information on the immune response of the patients, e.g., presence or absence of anti-toxin antibodies against the toxins expressed in the strains, then correlated with progression of lung dysfunction over time. The reviewer understands that this is not what the investigators proposed to study, but it is the study that the readers want to see, especially as readers of the journal “Toxins”.
Authors’ answer: We thank the reviewer for his/her recommendation and fully agree with him/her concerning the importance of host factors, which decisively influence the trajectory of S. aureus infections in humans. We included data concerning antibody levels against 19 virulence factors in our patients compared to healthy control subjects and also estimated an effect of antibody levels on FEV1% predicted:
Introduction, lines 82-84: “Furthermore, the host response to 19 of these virulence genes was assessed by measuring IgG-levels against the specific proteins by a bead-based flow cytometry technique (xMAP1; Luminex Corporation) as previously described [30,31].”
Materials and Methods, lines 175-177: “Anti-staphylococcal antibodies: IgG antibody levels were quantified applying a bead-based flow cytometry technique (xMAP ®; Luminex Corporation) as previously described [30,31]. For further information see Junge et al. [26].”
Results, lines 269-276 and Tables 1 and S4: “Complementary data concerning a humoral response towards S. aureus antigens:
- aureus toxin directed antibody levels in CF patients have been associated to the severity of exacerbations and decline in FEV1% [52]. Therefore, we measured IgG-levels against 19 virulence factors in serum samples of 182 subjects and compared them to antibody levels of 53 healthy adults with persistent nasal S. aureus colonization (Tab. S4). While IgG-levels against six antigens were significantly elevated in patients compared to controls (Tab.1), IgG-levels against seven virulence factors were significantly associated to a decline in FEV1% predicted (Tab.1).”
Discussion, lines 409-411: “The gradual loss of virulence genes could be beneficial for the bacterium concerning the evasion of host defenses, since many virulence factors are targets for the immune system as obvious by mounting of specific IgG against such virulence factors (Tab. 1) [61].”
Reviewer’s comment: Lines 189 - 192: While the loss of one gene every 5 years is statistically significant, one wonders if this is biologically significant. It seems unlikely as the patients as gene loss did not correlate with disease progression.
Authors’ answer: We understand the reviewer´s doubts concerning the relevance of this finding. However, we came to the conclusion that it indeed might be an interesting hint concerning the long-term virulence gene composition of S. aureus in CF patients. An analogous process of virulence gene loss in chronic infection has been described for P. aeruginosa in, which we included now in our revised manuscript:
Discussion, lines 408-411: Similar processes have also been proven for chronic P. aeruginosa infection in CF patients [61, 62]. The gradual loss of virulence genes could be beneficial for the bacterium concerning the evasion of host defenses, since many virulence factors are targets for the immune system as obvious by mounting of specific IgG against such virulence factors (Tab. 1) [61].
- Nguyen, D.; Singh, P.K. Evolving stealth: Genetic adaptation of Pseudomonas aeruginosa during cystic fibrosis infections. Proc. Natl. Acad. Sci. 2006, 103, 8305–8306 and
- Smith, E.E.; Buckley, D.G.; Wu, Z.; Saenphimmachak, C.; Hoffman, L.R.; D’Argenio, D.A.; Miller, S.I.; Ramsey, B.W.; Speert, D.P.; Moskowitz, S.M.; et al. Genetic adaptation by Pseudomonas aeruginosa to the airways of cystic fibrosis patients. Proc. Natl. Acad. Sci. 2006, 103, 8487–8492.
The gradual loss of virulence genes could be beneficial for the bacterium concerning the evasion of host defense, since many virulence factors are targets for the immune system. We discussed these implications in the revised version of the manuscript.
Furthermore, our analysis is limited concerning the virulence genes considered, which only represent a small fraction of the extensive S. aureus virulence genome. A hypothetical extrapolation of the virulence gene number considered would also increase the rate of longitudinal virulence gene loss. However, future research has to show whether the trend identified in our data holds up on the level of a more comprehensive virulence gene profile or the entire genome using whole genome sequencing.
Reviewer’s comment: Line 284: It appears that presence of seb in Figure 2b did show increased exacerbation. This gene has also been found in worse outcome with S. aureus bacteremia.
Authors’ answer: The seb showed a prevalence of 1,17 % among exacerbation patients and of 2,49 % among non-exacerbation patients. The difference was not significant. Without intention, we made a mistake in presenting the data and we would like to apologize for this mistake in Figure 2b, in which the prevalence of seb among exacerbation patients erroneously appeared elevated. This error has been corrected in the revised version of the manuscript.
Reviewer’s comment: Line 335: There are two other limitations that should be highlighted. First, the study of the bacteria relies solely upon gene presence and not gene expression. This is a major limitation as S. aureus gene expression in vivo has been emphasized in CF lung that does not correlate with gene presence (see Goerke et al. Infect Immun 68: 1304, 2000; Riquelme et al. Front Immunol 11: 385, 2020). Second, this study focuses upon the bacterium, but the host response is not included. Humans show variable responses to making antibodies to toxins, e.g., some women failed to make antibodies against tst, and they remained susceptible to toxic shock syndrome even after recovering from 1 episode.
Authors’ answer: We apologize for not mentioning the admittedly major limitation concerning the lack of proteomic data. We fully acknowledge the fact, that the mere presence of genes is far from being a sufficient condition for their actual translation into proteins and that the high levels of phenotypic diversity among S. aureus clones are not only due to genetical differences, but also result of differences in gene expression. We mention and discuss this limitation in the revised version of the manuscript. Furthermore, we now included data about the host response against most of the investigated virulence genes. Since for many virulence genes, CF patients mount higher antibody titers compared to healthy individuals, this is most likely due to expression of these factors during the in vivo growth of S. aureus within the airways of CF patients.
Discussion, lines 447-451: “Moreover, the analyses are solely based on genetic data. Complementary proteomic data would be desirable, since differences in gene expression contribute to the high level of phenotypic diversity among S. aureus clones [69] . Especially the diversity of the S. aureus exoproteome is considered to be relevantly shaped by gene regulation mechanisms [70, 71].”
- Wolf, C.; Kusch, H.; Monecke, S.; Albrecht, D.; Holtfreter, S.; von Eiff, C.; Petzl, W.; Rainard, P.; Bröker, B.M.; Engelmann, S. Genomic and proteomic characterization of Staphylococcus aureus mastitis isolates of bovine origin. Proteomics 2011, 11, 2491–2502.
- Ziebandt, A.K.; Kusch, H.; Degner, M.; Jaglitz, S.; Sibbald, M.J.J.B.; Arends, J.P.; Chlebowicz, M.A.; Albrecht, D.; Pantuček, R.; Doškar, J.; et al. Proteomics uncovers extreme heterogeneity in the Staphylococcus aureus exoproteome due to genomic plasticity and variant gene regulation. Proteomics 2010, 10, 1634–1644.
- Sibbald, M.J.J.B.; Ziebandt, A.K.; Engelmann, S.; Hecker, M.; de Jong, A.; Harmsen, H.J.M.; Raangs, G.C.; Stokroos, I.; Arends, J.P.; Dubois, J.Y.F.; et al. Mapping the Pathways to Staphylococcal Pathogenesis by Comparative Secretomics. Microbiol. Mol. Biol. Rev. 2006, 70, 755–788.
Minor issues:
Reviewer’s comment: Line 18: The sentence construction causes difficulty in understanding. Specifically, the phrase “hypothesis that rather the…” appears, but it does not work in English grammar and is very unclear…..This needs to be revised.
Authors’ answer: We apologize for this unclear sentence and rephrased it for the revised version of the manuscript.
Lines 25-26: “The results suggest that the phylogenetic background of clones defines S. aureus pathogenicity rather than the presence of specific virulence genes.”
Reviewer’s comment: Line 42: “infecting phenotype” is unclear and needs to be defined, although it has been used previously, it is a term that is not in wide use. An earlier paper should be referenced that discusses the different types of S. aureus in the CF lung (Goerke C et al. Environ Microbiol. 9 :3134, 2007)
Author´s answer: We replaced “infecting phenotype” with “infecting lifestyle” and gave an explanation for this. In this revised version of our manuscript, we also cited the recommended paper by Goerke et al.
Abstract, lines 29-30:”The longitudinal loss of virulence genes most likely reflects the adaptation process directed towards a persistent and colonizing, rather than infecting lifestyle.”
Introduction, lines 53-54: “The overall trend of S. aureus adaptation in CF airways is directed towards a persistent and colonizing, rather than an infecting lifestyle [6,12]”
Discussion, lines 406-408: “The finding is also consistent with the persistent and colonizing rather than infectious lifestyle, which S. aureus displays in CF patients [6].”
- Goerke, C.; Wolz, C. Adaptation of Staphylococcus aureus to the cystic fibrosis lung. Int. J. Med. Microbiol. 2010, 300, 520–525.
- Goerke, C.; Gressinger, M.; Endler, K.; Breitkopf, C.; Wardecki, K.; Stern, M.; Wolz, C.; Kahl, B.C. High phenotypic diversity in infecting but not in colonizing Staphylococcus aureus populations. Environ. Microbiol. 2007, 9, 3134–3142.
Reviewer’s comment: Line 56: The grammar in this sentence is difficult, making it hard to understand. This needs to be revised.
Author´s answer: We apologize for this unclear sentence and rephrased it for the revised version of the manuscript.
Introduction, lines 71-75: “Patients receiving intensive antibiotic therapy regimens more likely carried S. aureus isolates during persistence, which belonged to only one spa-type [7], while older patients carried isolates with related spa-types with mutations of the repeat region including deletions, duplications of repeats or point-mutations within repeats or dominant spa-types were cultured [7].”
- Westphal, C.; Görlich, D.; Kampmeier, S.; Herzog, S.; Braun, N.; Hitschke, C.; Mellmann, A.; Peters, G.; Kahl, B.C.; Junge., S.; et al. Antibiotic Treatment and Age Are Associated With Staphylococcus aureus Carriage Profiles During Persistence in the Airways of Cystic Fibrosis Patients. Front. Microbiol. 2020, 11.
Reviewer 2 Report
The manuscript “Staphylococcus aureus pathogenicity in cystic fibrosis patients results from an observational prospective multicenter study concerning virulence genes, phylogeny and horizontal gene transfer” deals with the evolutionary processes of S. aureus in a long-term occupation of CF patients. The study presents interesting (although not in principle new) results. The high number of isolates, patients and centers makes the study even more interesting.
The article is well structured and easy to read (unfortunately full of sloppy errors), see minor commends. The experiments and results are all comprehensible and from my point of view in order. The literary restoration is very extensive but does not contain important works (like Azarian et al 2019 or Ankrum 2017) or in some places there should be more than one quotation (e.g. CA-MRSA prevalence in CF). In the title and the first part of the manuscript the authors write about “horizontal gene transfer” but I do not find results or experiments about this objective.
Nevertheless, the manuscript can be put into a form that is suitable for publication.
minor commends
There are many errors in the manuscript regarding the nomenclature. S. aureus is not written in italics except for the introduction, as are P. aeruginosa and Burkholderia. P. aeruginosa must be written as Pseudomonas aeruginosa the first time it appears in the text!
Also genes must be written in italic (including the spa in spa typing!).
PVL is not the name of the PVL genes it is the name of the two component toxin (proteins), therefore it must not written in in small letters and italic.
Numbers smaller than 10 should be advertised, especially on page 2 (one Austrian CF center; two airway culture etc.)
The authors could put a list of the found spa types (and their number) in the supplementary data. The data can also be viewed in other papers but would be an experience for the readers. Which spa types have to be considered most prevalent (also numbers) but have to be written in the manuscript.
Author Response
Reviewer 2
Reviewer’s comment:The manuscript “Staphylococcus aureus pathogenicity in cystic fibrosis patients results from an observational prospective multicenter study concerning virulence genes, phylogeny and horizontal gene transfer” deals with the evolutionary processes of S. aureus in a long-term occupation of CF patients. The study presents interesting (although not in principle new) results. The high number of isolates, patients and centers makes the study even more interesting.
Authors’ answer: We would like to thank this reviewer for his/her overall positive evaluation of our manuscript.
Reviewer’s comment: The article is well structured and easy to read (unfortunately full of sloppy errors), see minor commends. The experiments and results are all comprehensible and from my point of view in order. The literary restoration is very extensive but does not contain important works (like Azarian et al 2019 or Ankrum 2017) or in some places there should be more than one quotation (e.g. CA-MRSA prevalence in CF).
Authors’ answer: We apologize for not mentioning the literature of Azarian et al. 2019 and Ankrum et al. 2017. In the revised version of our manuscript, we mentioned and discussed the work of these authors and also included more quotations about CA-MRSA prevalence in CF.
Introduction, lines 47-48:”Persistent S. aureus clones tend to form a genetically and particularly phenotypically heterogeneous population with S. aureus acquisition most likely outside the hospital with only minor transmission events [6-8].”
Introduction, lines 48-50: “The genetical diversity is fostered by mutations and horizontal gene transfer with persistence of dominant lineages and a continuous process of intrahost adaptation [6-9].”
Discussion, lines 413-418: “While Azarian et al. used WGS to investigate intra-host diversity of sequential MRSA isolates in four CF patients, the authors focused in their analysis, for which they used sequencing data of a mean of 21 isolates per patient from a mean period of 1,403 days, on the evidence of switched intergenic regions, allele frequency and single nucleotide polymorphisms but not the presence or absence of special virulence genes [9]. Therefore, it is not possible to compare their data with our analysis.”
- Goerke, C.; Wolz, C. Adaptation of Staphylococcus aureus to the cystic fibrosis lung. Int. J. Med. Microbiol. 2010, 300, 520–525.
- Westphal, C.; Görlich, D.; Kampmeier, S.; Herzog, S.; Braun, N.; Hitschke, C.; Mellmann, A.; Peters, G.; Kahl, B.C.; Junge., S.; et al. Antibiotic Treatment and Age Are Associated With Staphylococcus aureus Carriage Profiles During Persistence in the Airways of Cystic Fibrosis Patients. Front. Microbiol. 2020, 11.
- Ankrum, A.; Hall, B.G. Population dynamics of Staphylococcus aureus in cystic fibrosis patients to determine transmission events by use of whole-genome sequencing. J. Clin. Microbiol. 2017, 55, 2143–2152.
- Azarian, T.; Ridgway, J.P.; Yin, Z.; David, M.Z. Long-Term Intrahost Evolution of Methicillin Resistant Staphylococcus aureus Among Cystic Fibrosis Patients With Respiratory Carriage. Front. Genet. 2019, 10, 1–14.
Reviewer’s comment: In the title and the first part of the manuscript the authors write about “horizontal gene transfer” but I do not find results or experiments about this objective.
Nevertheless, the manuscript can be put into a form that is suitable for publication.
Authors’ answer: We reevaluated the passages of the manuscript concerning horizontal gene transfer and came to the conclusion that the reviewer has a point, since we did not measure horizontal gene transfer directly, but used the surrogate parameter “virulence gene plasticity” to estimate the potential for horizontal gene transfer. We consider virulence gene plasticity as a valid surrogate parameter for the potential for horizontal gene transfer, since under conditions, which have been proven to foster horizontal gene transfer (e.g high bacterial density, presence of related clones, high environmental stress) our data show increased virulence gene plasticity.
We apologize for the confusing use of concepts in this regard and changed “horizontal gene transfer” in the title into “gene plasticity”.
Minor issues:
Reviewer’s comment There are many errors in the manuscript regarding the nomenclature. S. aureus is not written in italics except for the introduction, as are P. aeruginosa and Burkholderia. P. aeruginosa must be written as Pseudomonas aeruginosa the first time it appears in the text!
Also genes must be written in italic (including the spa in spa typing!).
PVL is not the name of the PVL genes it is the name of the two component toxin (proteins), therefore it must not written in in small letters and italic.
Numbers smaller than 10 should be advertised, especially on page 2 (one Austrian CF center; two airway culture etc.)
Authors’ answer: We apologize for the formal mistakes and errors regarding nomenclature. They are corrected in the revised version of the manuscript.
Reviewer’s comment: The authors could put a list of the found spa types (and their number) in the supplementary data. The data can also be viewed in other papers, but would be an experience for the readers. Which spa types have to be considered most prevalent (also numbers) but have to be written in the manuscript.
Authors’ answer: We thank the reviewer for his/her recommendation and included the required material into the supplementary material as Tables S2 and S3.
Round 2
Reviewer 1 Report
The investigators have responded very well to all questions. The new data make the data much stronger. An error concerning SEB was corrected. The grammar errors have been corrected.